# A Novel Peptide, CK2.3, Improved Bone Formation in Ovariectomized Sprague Dawley Rats

**DOI:** 10.3390/ijms21144874

**Published:** 2020-07-10

**Authors:** Linda Sequeira, John Nguyen, Liyun Wang, Anja Nohe

**Affiliations:** 1Department of Biological Sciences, University of Delaware, Newark, DE 19716, USA; seq@udel.edu (L.S.); njohn@udel.edu (J.N.); 2Department of Mechanical Engineering, University of Delaware, Newark, DE 19716, USA; lywang@udel.edu

**Keywords:** osteoporosis, bone formation, osteoblast differentiation, BMP2, casein kinase 2, CK2.3

## Abstract

Osteoporosis is a bone disease that has no definite cure. Current treatments for osteoporosis are divided into two categories: anti-resorptive and anabolic. However, these treatments are not perfect and have considerable risks. In addition, bone quality often declines over time with these treatments. We designed a peptide, CK2.3, that has both anabolic and anti-resorptive effects on bone. We reported that CK2.3 induced osteoblastic mineralization, promoted bone formation, and suppressed osteoclastogenesis in vivo. The effect of CK2.3 to rescue an osteoporosis phenotype model has never been shown. In this study, we demonstrated the effect of CK2.3 in ovariectomized rats, a standard model of osteoporosis. We systemically injected CK2.3 at 2.3 µg/kg each day for five consecutive days. Micro-computed tomography indicated that CK2.3 increased bone mineral density, (bone volume/tissue volume) BV/TV and (trabecular number) TbN, and decreased (trabecular space) TbSp in the femoral head. Similarly, single photon absorptiometry showed that treatment with CK2.3 increased bone mineral density in the lumbar spine and the pelvis. Additionally, we observed increased femoral shaft stiffness with ovariectomized rats treated with CK2.3. We also detected no significant changes in the weight of organs such as the heart, lung, liver, kidney, and spleen. An advantage of CK2.3 over current treatments was that it not only promoted bone formation but also improved fracture resistance. In conclusion, we demonstrated CK2.3 as a new anabolic treatment for osteoporosis.

## 1. Introduction

Bone remodeling is a dynamic process that is characterized by the balance between osteoclast and osteoblast activity. About 10% of the skeletal bone is being resorbed and replaced by the action of osteoclasts and osteoblasts, respectively, annually [1]. However, when osteoblast activity can no longer keep up with osteoclast activity, it will eventually lead to a condition called osteoporosis (OP), a bone disease characterized by the loss of bone mineral density. OP is often associated with aging. The effect of age on bone remodeling has been well studied. In these studies, osteoblastogenesis declines whereas osteoclastogenesis increases with age [2,3,4]. In 2010, It was estimated that 54% of people in the United States over the age of 50 were diagnosed with OP or osteopenia [5]. Over USD 17 billion was spent for hospitalization admission on OP-related fractures in 2015, and it is expected to exceed USD 32 billion by 2025 in the United States [6].

There are two main classes of OP treatments: anti-resorptive and anabolic. Bisphosphonates, denosumab (RANK ligand inhibitor), calcitonin, estrogen (hormone therapy), selective estrogen receptor modulators (SERMs) are anti-resorptive treatments. Bisphosphonates are stable derivatives of inorganic pyrophosphates (PPi) that are capable of binding to hydroxyapatite crystals in the bone matrix. When they are taken in by osteoclasts, they induce apoptosis [7]. Treatment with bisphosphonates, however, is also associated with increased accumulation of micro-fractures [8,9]. Denosumab is a human RANK ligand monoclonal antibody that inhibits osteoclast formation, function, and survival by preventing the binding of RANK ligand to RANK on osteoclast surface [10]. Denosumab also inhibits the interaction of RANK ligand and RANK on the surface of immune cells such as T lymphocytes, B cells and dendritic cells, thus, increasing the risk of serious infection [11,12]. Calcitonin binds exclusively to osteoclasts in bone and causes suppression of bone resorption [13]. However, there is a lack of data to show calcitonin reduces non-vertebral fractures [6]. Activation of estrogen receptor by estrogen or SERMs reduces osteoclast differentiation and activity, while inducing osteoclast apoptosis [14]. However, overall health risks including stroke, cardiovascular event, venous thromboembolism may exceed the health benefits of estrogen and SERMs [6,15]. On the other hand, parathyroid hormones (PTH) and sclerostin monoclonal antibody are anabolic treatments. PTH analogs have been used for a long time as bone anabolic agents [16,17,18]. However, osteoblastogenesis by PTH also produces RANK ligand and M-CSF, the two important factors for osteoclast formation and activity. Elevated PTH can also cause increased bone resorption and high levels of blood calcium [18]. Sclerostin monoclonal antibody was approved as a new anabolic treatment for OP [19]. Sclerostin monoclonal antibody inhibits the function of sclerostin to promote osteoblast formation and mineralization via Wnt/B-catenin signaling pathway [20]. However, there is a lack of data about long-term safety of the sclerostin antibody. Although, it is reported a 30% increased risk of cardiovascular event with Romosozumab (sclerostin monoclonal antibody) over alendronate (a bisphosphonate derivative) [19].

Bone morphogenetic protein 2 (BMP2) became a potential new treatment for OP in 2002. BMP2 is a member of transforming growth factor beta (TGF-B) superfamily. BMP2 signals through bone morphogenetic protein receptor type 1a (BMPRIa) to exert osteogenesis as well as cell growth, differentiation, apoptosis, embryogenesis and development [21]. Osteogenic effects of BMP2 are demonstrated through its direct effect on promoting osteoblastogenesis and osteoclastogenesis [22,23,24,25]. Our previous study showed that casein kinase 2 (CK2) was a negative regulator of BMPRIa activity, and upon stimulation with BMP2 CK2 was released from BMPRIa allowing the activation of the SMAD signaling downstream of BMPRIa [26]. We designed a novel peptide named CK2.3 that mimicked the BMPRIa binding site (aa 213-217) for CK2. CK2.3 bound to CK2 to block the interaction between CK2 and BMPRIa [26]. CK2.3 demonstrated anabolic effect on bone by promoting mineralization in primary calvarial osteoblasts and bone marrow stromal cells (BMSC) [27]. Furthermore, in a study with low bone mineral density (BMD) 6-month-old retired breeder mice, CK2.3 was shown to promote mineral apposition rate, increased trabecular BMD of femurs, and increased lumbar spine BMD [28]. Unexpectedly, CK2.3 also improved femoral shaft bending stiffness after four weeks of treatment suggesting that CK2.3 promoted fracture resistance [28]. Our studies also found that CK2.3 exhibited anti-resorptive effects as seen via inhibition of osteoclast activity and differentiation [27,28], making CK2.3 ideal for long-term treatment of OP. We also demonstrated that myoblast C2C12 cells with BMPRIa mutants that lack a CK2 binding site were shown to promote mineralization, corroborating our findings that CK2.3 increases osteogenesis by blocking CK2 binding [29]. The mineralization, however, was suppressed with mitogen-activated protein kinase kinase (MEK) inhibitor PD98059, but not with p38 and mammalian target of rapamycin (mTOR) inhibitors, SB203580 and rapamycin, respectively [29]. Elevated extracellular signal-regulated kinase (ERK) activation was also observed in the femurs of CK2.3-injected 8-week-old mice [27]. This suggests that CK2.3 exerted its anabolic effect through the MEK/ERK signaling pathway. The underlying mechanism of the anti-resorptive effect of CK2.3, such as inhibition of osteoclastogenesis, is still unknown.

In this study, we demonstrated the effect of CK2.3 in an osteoporotic animal model, the ovariectomized (OVX) rats. OVX rats are an excellent model which emulate the estrogen depletion that leads to OP in post-menopausal women [30,31,32]. The OVX mouse, however, is used less frequently due to a deficiency of data validating it as an appropriate osteoporotic model [31]. Furthermore, ovariectomy-induced bone loss was reportedly different among inbred strain of mice [33,34,35]. On the other hand, ovariectomy-induced bone loss in outbred Sprague Dawley rats was validated by the increase of osteoclast surface and decrease of bone volume [36,37,38]. Additionally, according to Charles River (our vendor), they have about 97.2% of successful rate of complete removal of the ovary. Only in 2.8% of cases are there are remnants of the ovary, which may experience a partial regeneration. Overall, the OVX rat model is most commonly used to study a postmenopausal OP model [32]. A hypothesis of this study was that systemic injection of CK2.3 increased the BMD and bone architecture in OVX rats without influencing internal organs. CK2.3 was systemically injected into the animals once a day for five days consecutively. The concentration of CK2.3 was 2.3 µg/kg as previously reported [28]. Micro-computed tomography and single photon absorptiometry indicated that CK2.3 increased trabecular BMD in the femoral head and lumbar spine, respectively. Micro-computed tomography also indicated that CK2.3 improved trabecular architecture of femoral heads such as increased bone volume/tissue volume (BV/TV) and trabecular number (TbN) and decreased trabecular space (TbSp). A concern about CK2.3 was that it might have an effect on internal organs. However, the rats’ organs (liver, heart, lung, spleen, kidney) weights were unchanged, indicating that CK2.3 was a non-toxic peptide. Furthermore, three-point bending test showed that CK2.3 enhanced femoral shaft bending stiffness suggesting CK2.3 promotes fracture resistance. The findings in this study continued to demonstrate that CK2.3 was a promising new treatment for OP.

## 2. Results

### 2.1. CK2.3 Increases Trabecular Bone Architecture of the Femoral Head of OVX Rats

We first investigated the effect of CK2.3 on bone architecture in OVX rats. Micro-CT is a high-resolution imaging technique that is capable of analyzing bone structure. BV/TV represents the amount of mineralized bone at a given volume of interest. Trabeculae are the spongy network of bone. TbN represents the average number of trabeculae per unit length (1/mm). TbSp represents the distance between trabeculae (mm). Analysis of trabecular bone architecture of the femoral head by micro-CT (Figure 1) showed that CK2.3-injected OVX rats had higher BV/TV at week 4 and week 12, 60.63% vs. 46.22% and 62.90% vs. 42.51%, respectively (Figure 1A,D), reduced TbSp at week 4 and week 12, 0.11 mm vs. 0.14 mm and 0.10 mm vs. 0.15 mm, respectively (Figure 1B,E), and higher TbN at week 12, 4.50 mm^−1^ vs. 3.89 mm^−1^, to that of PBS-injected OVX rats (Figure 1F). In addition, CK2.3-injected OVX rats recovered trabecular bone architecture to a similar level of PBS-injected SHAM rats (Figure 1).

### 2.2. CK2.3 Recovered Trabecular BMD in the Femoral Head of OVX Rats

BMD is the measure of the amount of minerals (calcium and phosphorous) in a given volume of interest. Low BMD was the hallmark of OP, thus, we investigated if CK2.3 could improve the BMD in OVX rats. Baseline trabecular BMD measurement of the femoral head at week 0 confirmed that OVX rats had lower trabecular BMD than SHAM rats, 0.21 g/cm^3^ vs. 0.26 g/cm^3^, (Figure 2A). However, CK2.3-injected OVX rats recovered their trabecular BMD, and it was significantly higher to that of PBS-injected OVX rats at week 4 and week 12 (0.24 g/cm^3^ vs. 0.20 g/cm^3^ and 0.23 g/cm^3^ vs. 0.19 g/cm^3^, respectively) (Figure 2B,C). Interestingly, although CK2.3-injected OVX rats still had significantly lower trabecular BMD than PBS-injected SHAM rats at week 4 (Figure 2B), CK2.3-injected OVX rats recovered their trabecular BMD to a similar level as PBS-injected SHAM rats at week 12 (Figure 2C).

### 2.3. CK2.3 Increased Lumbar Spine and Pelvis Trabecular BMD of OVX Rats

In addition to hip fractures, spinal and pelvic fractures are also a concern to osteoporotic patients. Therefore, we examined the effect of ovariectomy on the rat spine and pelvis. Then, we tested whether CK2.3 could rescue the condition of the lumbar spine and pelvis. Trabecular BMD of the pelvis and lumbar spines were determined by SPA and the L4 lumbar spine alone by micro-CT (Figure 3). SHAM rats and non-treated OVX rats had similar trabecular BMD in the pelvis but significantly different BMD in the lumbar spine. Treatment of OVX rats with CK2.3 showed significant improvement over non-treated OVX rats in trabecular BMD of both the pelvis and lumbar spine.

### 2.4. CK2.3 Increased Stiffness of Femoral Shaft of OVX Rats

Although, current treatments available for OP can improve bone formation, they do not often improve fracture resistance. The three-point bending test is a technique that can be used to evaluate the mechanical strength of bone. Force is applied until the breaking point is reached, and the bone breaks. From this, factors such as stiffness and fracture resistance can be calculated. Thus, we calculated the bending stiffness of femurs to examine the effect of CK2.3 on fracture resistance. Analyzing femoral shaft stiffness by three-point bending test showed that PBS-injected OVX rats had the lowest bending stiffness of femoral shaft (Figure 4). However, CK2.3 was able to restore the bending stiffness of the femoral shaft of OVX rats at week 12 (Figure 3).

### 2.5. CK2.3 Did Not Affect Organs of OVX Rats

Finally, we examined the possible effect of CK2.3 on the rat internal organs. Organs such as the lung, heart, spleen, liver, and kidney were weighed to determine their mass (Figure 5). Our analysis showed that CK2.3 treatment neither increased nor decreased the mass of these organs in OVX rats. Similarly, we observed no changes in organ morphology with CK2.3 treatment.

## 3. Discussion

OP is an age-dependent disease that has no cure. Current treatments are only alleviating symptoms, but do not abolish the disorder. They often have negative impacts on patients’ lives. Our discovery of a novel peptide, CK2.3, however, could potentially improve the patients’ quality of living. From our previous studies, we designed CK2.3 as a peptide to block the interaction between CK2 and BMPRIa [26]. BMP2 is emerging as a new therapeutic treatment for bone fracture because of its essential involvement in osteoblastogenesis and bone formation [22,39,40,41,42]. Nevertheless, other negative impacts of BMP2, including direct enhancement of osteoclastogenesis, prevented BMP2 from becoming an ideal treatment for OP [25,43,44]. However, the BMP2 signaling pathway may still be used to treat OP. We showed that CK2 was a binding protein on BMPRIa and a negative regulator of the BMP2 signaling pathway [26,45]. By blocking the interaction between CK2 and BMPRIa with CK2.3 peptide, we demonstrated that it could stimulate the downstream canonical Smad pathway even in the absence of the BMP2 ligand [26]. CK2.3 promoted mineralization and serum level of osteoblastic markers such as alkaline phosphatase (ALP) and osteocalcin (OC) in vivo [27,45]. In addition, low BMD 6-month-old retired breeder mice injected with CK2.3 showed enhancement in trabecular BMD of the femur and lumbar spine, and bone mineral apposition rate [28]. Interestingly, CK2.3 also suppressed osteoclast activity and osteoclastogenesis and decreased serum level of TRACP 5b, an osteoclastic marker [27,28]. Surprisingly, CK2.3 was also found to enhance femoral shaft stiffness suggesting that CK2.3 improved fracture resistance [28].

Our previous study of 6-month-old retired breeder mice was based on the low BMD of the mice due to age and giving birth multiple times. This model did not reflect the change in BMD due to age and estrogen deficiency as observed in post-menopausal women who are most commonly affected by OP. Therefore, to further confirm the therapeutic potential of CK2.3, we systemically injected CK2.3 into OVX rats, a standard model to study OP. In the early period after OVX, increased osteoclast surface together with increased osteoblast surface, mineralized surface and bone formation rate are detected in first lumbar vertebral body [36]. However, around 30 days post-OVX, osteoblast surface, mineralized surface and bone formation rate begin to fall [36]. Therefore, treatments started 4 weeks post-OVX in order to match the bone loss period. According to the Charles River Laboratory (our vendor) the successful rate of complete removal of the ovary is 97.2%. Of 2.8% cases with a leftover ovary, very few cases might experience a slight regeneration of the ovary. Additionally, it is reported that OVX rats have an average of 15% and 20% loss of femoral BMD at week 4-12 and after week 12 post OVX, respectively [46]. In our study, OVX rats had about 20% decreased BMD in the femur compared to SHAM rats at week 4-16 post OVX. Other researchers have also reported similar loss of femoral BMD (about 20% loss at week 4-20 post OVX) in their OVX rat model [47,48,49,50,51]. Therefore, we did not investigate the atrophy of the uterus to confirm the successful rate of the OVX surgery. For this study, we used the same concentration (2.3 µg/kg) as reported in the previous 6-month-old retired breeder mice study [28]. We observed that CK2.3 improved trabecular BMD of the femoral head in OVX rats (Figure 2). Additionally, micro-CT analysis of femoral bones indicated an increase in BV/TV and TbN, and a decrease in TbS for OVX rats with CK2.3-injection compared to the PBS injected OVX rats (Figure 1). Moreover, CK2.3 also improved lumbar spine trabecular BMD in CK2.3-injected OVX rats (Figure 3B,C). The time periods week 4 and week 12, which were chosen in this study, were 56 and 112 days post-OVX, respectively. During these time periods, mineralized surface and bone formation rate are reported to decreased [36]. Femoral head BMD, sacrum spine BMD and lumbar spine BMD were shown to decreased in PBS control but increased in CK2.3 treatment (Figure 2 and Figure 3) showing the effectiveness of CK2.3. These results showed that the effect of CK2.3 was similar to those of other bone anabolic drugs such as PTH and sclerostin monoclonal antibody [16,17,20]. Due to be a relatively new treatment, long-term safety of sclerostin monoclonal antibody has not been validated. However, it is reported that elevated Wnt/b-catenin signaling pathway, activated by inhibition of sclerostin by Scl-ab, is associated with osteosarcoma [52]. Similarly, PTH treatment was showed to reduce serum sclerostin levels in postmenopausal women that may also lead to osteosarcoma [53]. CK2.3, on the other hand, showed that at higher concentration (three times higher, 6.9 µg/kg) did not result in increased osteoclast formation [28]. In this study, we also demonstrated the efficacy of CK2.3 at low dosage (2.3 µg/kg once per day for 5 days). It is also noteworthy that CK2.3 was used at a lower concentration and shorter duration of treatment than those of PTHrP and sclerostin monoclonal antibody (30 µg/kg once per day for 4 weeks and 25 mg/kg twice per week for 5 weeks, respectively) [54,55].

Osteoporotic fractures often occur at the femoral neck and vertebrate. A small amount of osteoporotic fractures also occurs at the pelvis. Nevertheless, pelvic fractures, like any other fractures at the femoral neck or vertebrate, can also have severe consequences such as: immobility, impair quality of life, and high rate of mortality. Today, the cases of pelvic fractures have been on the rise. From 1991 to 2007, the incidences of pelvic fractures increased by 65% [56]. Thus, attention should be paid more to pelvic fractures rather than femoral neck and vertebral fractures. The escalating number of pelvic fractures shows that current treatments of OP have not been effective to protect patients. In this study, treatment with CK2.3 significantly increased trabecular BMD in the pelvis of OVX rats (Figure 3A). This result also further showed that CK2.3 improved BMD of the skeletal system as a whole rather than just at several particular sites such as femurs and lumbar spine.

Bone remodeling is the process when bone formation is coupled with bone resorption and work in a sequence of activation-resorption-formation [32,57]. Bone modeling, on the other hand, is the process when bone formation and bone resorption work independently from each other [32,57]. Bone modeling defines skeletal development and bone growth; thus, it is a predominantly early development activity. The transition from modeling to remodeling occurs later. To conduct a research on a potential treatment for osteoporosis, the bone sites and the age of the animal must be such that remodeling is the predominant activity [32]. In this study, the rats were 4-months-old when the ovary was removed. 4 weeks post OVX, they were 5-months-old when treatments were initiated. At week 4 and week 12 post treatments, they were about 6-months-old and 8-months-old, respectively. The transition time from modeling to remodeling of the proximal tibial metaphysis in the cancellous bone and endocortical bone is 6–9 months and 9–12 months, respectively [32]. Due to the age of the rats, we determined that using tibiae for this study would not be appropriate. Additionally, the effect of osteoporosis treatments on improving BMD has shown to be similar in femur and tibia [58,59,60,61]

Current anti-resorptive drugs such as bisphosphonates, although effective, also lead to an increase in atypical fractures [62]. Just as in the previous study with 6-month-old female retired breeder mice, we observed an improvement in femoral shaft bending stiffness of CK2.3 injected OVX rats (Figure 4). Thus, it suggested that CK2.3 promoted bone fracture resistance—an advantage over other currently available anti-resorptive treatments on the market.

Furthermore, this study suggested that CK2.3 did not have a negative impact on the OVX rats as no changes in morphology and tissue weight of internal organs were detected (Figure 5). It is shown that organ weight can be used as an assessment in toxicity studies [63]. Although future studies will need to look at potential toxicity of CK2.3 in more details, nevertheless, the unchanging in organ weight in this study can be used as a reference in future studies of CK2.3. From this study, CK2.3 has shown to be a promising treatment for OP that could promote BMD, while also maintaining bone architecture. Whether or not CK2.3 enhanced femoral shaft stiffness due to CK2.3′s ability to mediate microdamage repair is worth investigating. This will indicate whether or not CK2.3 affects the bone remodeling process. Furthermore, the precise mechanism of CK2.3 has yet been elucidated. Understanding how CK2.3 works will add new information that will advance the field of OP research.

In conclusion, our study demonstrated CK2.3 is a promising treatment for OP. We developed CK2.3 as a new anabolic treatment for OP. We demonstrated that CK2.3 not only promotes BMD in an osteoporotic model (OVX rats) but also improves bone fracture resistance. CK2.3 could offer patients an alternative treatment option for a better quality of life.

## 4. Materials and Methods

### 4.1. Design of Blocking Peptide CK2.3

CK2.3 was designed as previously described (15). Briefly, using the Prosite database search a possible CK2 phosphorylation site on BMPRIa was predicted at amino acid 213-217 (SLKD). The peptide was designed with a CK2 phosphorylation site (SLKD) and an Antennapedia homeodomain signal sequence for cellular uptake [64]. Each site of the peptide was flanked with several amino acids, creating a total length of 29 amino acids.

### 4.2. Rat Injection and Live Micro-Computed Tomography of the Femoral Head

Fourteen female OVX and seven placebo (SHAM) operated Sprague Dawley rats, 15 weeks of age (105 days) were purchased from Charles River Laboratory (QTE# 20150209, Wilmington, MA, USA). The animals were housed and fed in the animal facility of Office of Laboratory Animal Medicine, University of Delaware (Newark, DE, US). Upon arrival, they were housed (two per ventilated cage) in a specific pathogen-free room. The rats were allowed free access to tap water and commercially standard rodent food.

Four weeks after arrival, baseline bone mineral densities (BMD) of the trabecular left femur were measured using the Bruker SkyScan 1276 X-ray micro-computed tomography (micro-CT) (Bruker, Kontich, Belgium). Two days later, the rats’ tail veins were injected with 50 µl of either phosphate buffered saline (PBS) or 2.3 µg/kg of CK2.3. The rats were injected as follows: 7 SHAM rats and 7 OVX rats were injected with PBS and 7 OVX rats received the CK2.3 peptide. One injection per day for all rats continued for five consecutive days. Four weeks after the last injection or nine weeks after OVX, the rats were once again subjected to micro-CT to determine the bone architecture of the left femur such as BMD, BV/TV, TbSp, and TbN. 

At the conclusion of the week four BMD readings, the rats were returned to the Animal Facility and eight weeks later, the rats were subjected to micro-CT to assess the bone architecture of the left femur. At this point, the rats were about 33 weeks of age. Then, the rats were sacrificed using compressed CO2 gas at an initial flow rate of approximately 2 L/minute until unconsciousness was achieved, and then the rate was increased to 10 L/minute whilst observing faded eye color and cessation of respiration. This was followed by a bilateral thoracotomy.

Animal protocol (AUP#1194) use was registered and approved by the Institutional Animal Care and Use Committee of the University of Delaware (5 Sep 2019).

### 4.3. Micro-Computed Tomography (micro-CT)

Micro-CT was conducted by placing the rats in an appropriately sized cassette equipped with three connectors: anesthetic gas (isofluorane), 3-way ECG, and a temperature sensor. The physiological monitoring (breathing, temperature, and ECG) was connected throughout each scan using the integrated aforementioned sensors. Video monitoring for real time movement was conducted using a 5Mp CMOS camera.

For reconstruction of the image parameters, post-alignment, beam hardening, ring artifacts reduction, and smoothing were adjusted to get an optimum image quality. The reconstructed dataset was loaded for 3D viewing using the DataViewer v. 1.5.6.2 (Bruker, Kontich, Belgium). This program produces 3 orthogonal planes, x-z or coronal view, x-y coronal view, and x-z sagittal view. To view transverse sections of the desired trabecular bone, we focused on a volume of interest (VOI) and saved all sagittal views.

The analysis was done in CTAn scanned at 20 microns to determine the BMD of the region of interest (ROI).

### 4.4. Single Photon Absorptiometry (SPA) of Rat Pelvis and Lumbar Spine and Calculation of BMD

BMD of the lumbar spine and pelvis were measured by SPA, as the X-ray penetrates through the sample in a single photon ray and is reflected onto a detector [65]. Single view radiographs were imaged of each lumbar spine on a ScanX Duo digital imaging system (Air Techniques, Melville, NY, USA). Exposure time was 0.40 sec. The kilovoltage peak (kVp) was 60. The milliampere-second (mAs) was 2.5.

The pixel intensity (PI) was calculated for each radiograph in ImageJ v 1.8.0 (NIH, Bethesda, MD, USA) to estimate BMD. PI is a measurement of a grey level value on a scale of 0 (black) to 255 (white) and has been shown to correspond with bone density (mineralized) or BMD in several other studies [66,67,68].

### 4.5. Three-Point Bending Test

Femur stiffness was analyzed by three-point bending test using an Instron 5848 Micro Tester (Instron, Norwood, MA, USA). The mid-shaft of the femur was compressed in the anterior-posterior direction with a lower support span of 20.0 mm and a loading rate of 0.1mm/min. The instrument was set to stop when the bone was fractured. The stiffness was defined as the slope of the linear region of the displacement vs. force graph.

### 4.6. Statistical Analysis

In order to reduce false positive results due to CK2.3’s negative effect on OVX rats, statistical significance was determined by using one-way ANOVA followed by Fisher’s least significant difference post-hoc test, with one-tail α < 0.1

## Figures and Tables

**Figure 1 ijms-21-04874-f001:**
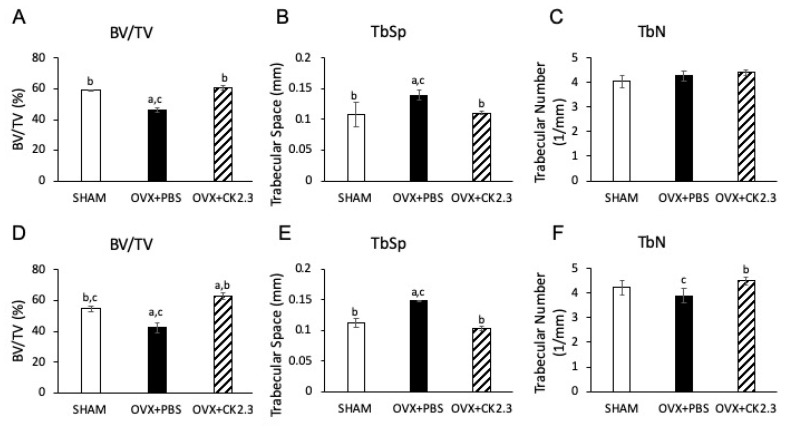
Micro-CT analysis of rat femoral head at week 4 (**A**–**C**) and week 12 (**D**–**F**). OVX rats injected with CK2.3 showed improvements in bone architecture. (**A**,**D**) CK2.3 increased BV/TV ratio; (**B**,**E**) CK2.3 decreased TbSp trabecular space; (**C**,**F**) CK2.3 enhanced TbN trabecular number only at week 12 (**F**) but not at week 4 (**C**). (a) Indicated a significant difference over SHAM, (b) indicated a significant difference over OVX+PBS, and (c) indicated a significant difference over OVX+CK2.3.

**Figure 2 ijms-21-04874-f002:**
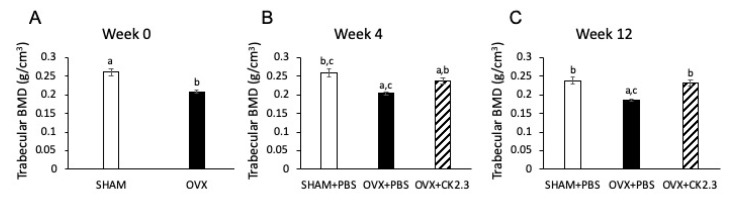
CK2.3 increased trabecular BMD femoral head of OVX rats. (**A**) OVX rats had lower baseline (week 0) BMD than SHAM rats; (**B**,**C**) Treatment with CK2.3 show a significant improvement in BMD in the femur of OVX rats in week 4 (**B)** and week 12 (**C)**. (a) indicated a significant difference over SHAM, (b) indicated a significant difference over OVX or OVX+PBS, (c) indicated a significant difference over OVX+CK2.3.

**Figure 3 ijms-21-04874-f003:**
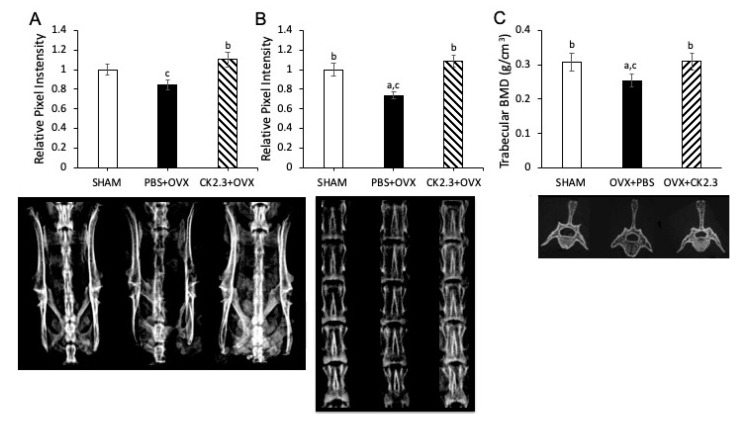
CK2.3 increased trabecular BMD of the spine of OVX rat at week 12. (**A**) CK2.3 improved trabecular BMD of sacrum spine measured by SPA; (**B**) CK2.3 increased trabecular BMD of lumbar spine measured by SPA; (**C**) CK2.3 increased trabecular BMD of L4 lumbar spine measured by micro-CT. (a) indicated a significant difference over SHAM, (b) indicated a significant difference over OVX+PBS, (c) indicated a significant difference over OVX+CK2.3.

**Figure 4 ijms-21-04874-f004:**
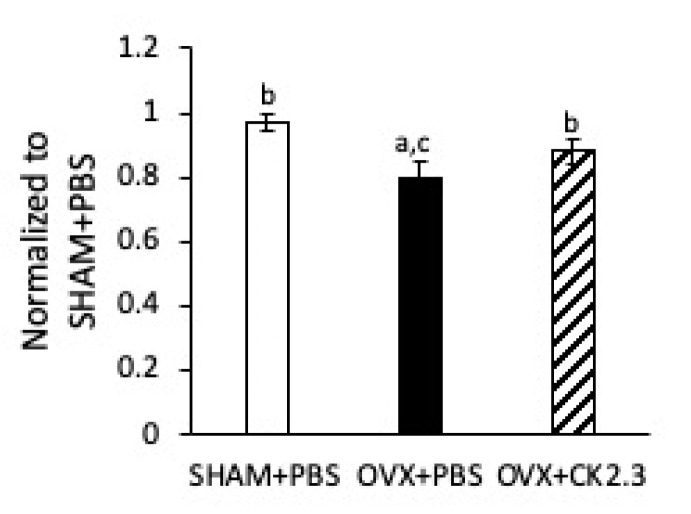
CK2.3 improved femoral shaft stiffness of OVX rats. Both left and right femurs were subjected to 3-point bending test. CK2.3-injected OVX rats’ femur had significantly higher stiffness than PBS-injected OVX rats. (a) Indicated a significant difference over SHAM, (b) indicated a significant difference over OVX+PBS, and (c) indicated a significant difference over OVX+CK2.3.

**Figure 5 ijms-21-04874-f005:**
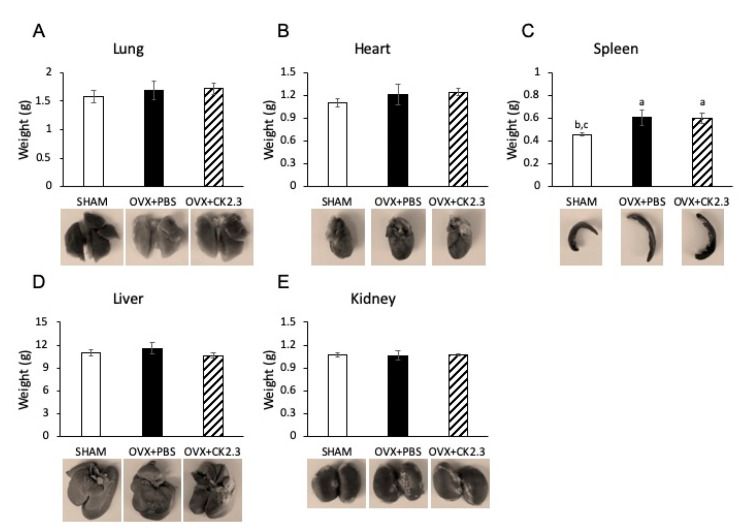
CK2.3 did not affect lung, heart, liver, spleen, and kidney of OVX rats. Mass of (**A**) lung, (**B**) heart, (**C**) spleen, (**D**) liver, and (**E**) kidney was similar in CK2.3-injected OVX rats to PBS-injected OVX rats. (a) Indicated a significant difference over SHAM, (b) indicated a significant difference over OVX+PBS, and (c) indicated a significant difference over OVX+CK2.3.

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
