# Peer review of "A Novel Peptide, CK2.3, Improved Bone Formation in Ovariectomized Sprague Dawley Rats"

_ijms, 2020, doi:10.3390/ijms21144874_

Round 1

Reviewer 1 Report

In the present study, CK2.3 improved bone formation in ovariectomized Sprague Dawley rats. I have few comments.

Major points

1. Fig5, about the side effect to each organ.

As well as the weight, please show a normal organ by HE staining.

2. If you see influence on osseous tissue, e.g., please show a result of μCT, TRAP and von kossa staining of the proximal tibiae

Author Response

We appreciated the opportunity to revise the manuscript for reconsideration. We also want to thank the reviewer for the thoughtful and thoroughly comments on the manuscript. We took every comment seriously and revised the manuscript accordingly. The following content is a point-by-point revision based on reviewer’ comments and suggestions

Reviewer 1
Major points
1. Fig 5, about the side effect to each organ. As well as the weight, please show a normal organ by HE staining.
Thank you for the comment, and we understand the concern of the reviewer about this issue. We initially decided not to perform HE staining of organs because based on our previous studies of the CK2.3 peptide, we did not detect any health issues with the animals (1-5). Additionally, we did not observe any abnormalities with the organs after isolation (Figure 5). Furthermore, it is shown that evaluation of organ weight can be used as an assessment for toxicity studies (6). Thus, based on the unchanging in weight, we extrapolated that CK2.3 didn’t negatively affect the organs.

2. If you see influence on osseous tissue, e.g., please show a result of mCT, TRAP and von kossa staining of the proximal tibiae.
We understand the point the reviewer made. Our initial decision not to perform mCT, TRAP and von Kossa staining of the proximal tibiae was because we showed previously the effect of CK2.3 inhibited osteoclastogenesis and promoted osteoblastogenesis in vitro and in vivo (1-5, 7-9). Furthermore, we demonstrated the effect of CK2.3 on primary cells isolated from bones donated by patients (8). Thus, we opted out of performing TRAP and von Kossa staining in this study. Additionally, the effect of osteoporosis treatments on improving BMD has shown to be similar in femur and tibia (10-13). Therefore, we didn’t study the effect of CK2.3 on tibia. Although, we didn’t perform mCT on the proximal tibiae, we showed the results of mCT of femoral head.

References
1. Bragdon B, Thinakaran S, Moseychuk O, King D, Young K, Litchfield DW, et al. Casein kinase 2 beta-subunit is a regulator of bone morphogenetic protein 2 signaling. Biophys J. 2010;99(3):897-904. Epub 2010/08/05. doi: 10.1016/j.bpj.2010.04.070. PubMed PMID: 20682268; PubMed Central PMCID: PMCPMC2913199.
2. Bragdon B, Thinakaran S, Moseychuk O, Gurski L, Bonor J, Price C, et al. Casein kinase 2 regulates in vivo bone formation through its interaction with bone morphogenetic protein receptor type Ia. Bone. 2011;49(5):944-54. Epub 2011/07/19. doi: 10.1016/j.bone.2011.06.037. PubMed PMID: 21763800.
3. Bragdon B, Bonor J, Shultz KL, Beamer WG, Rosen CJ, Nohe A. Bone morphogenetic protein receptor type Ia localization causes increased BMP2 signaling in mice exhibiting increased peak bone mass phenotype. J Cell Physiol. 2012;227(7):2870-9. Epub 2011/12/16. doi: 10.1002/jcp.23028. PubMed PMID: 22170575; PubMed Central PMCID: PMCPMC3309108.
4. Akkiraju H, Bonor J, Olli K, Bowen C, Bragdon B, Coombs H, et al. Systemic injection of CK2.3, a novel peptide acting downstream of bone morphogenetic protein receptor BMPRIa, leads to increased trabecular bone mass. J Orthop Res. 2015;33(2):208-15. doi: 10.1002/jor.22752. PubMed PMID: 25331517; PubMed Central PMCID: PMCPMC4304894.
5. Nguyen J, Weidner H, Schell LM, Sequeira L, Kabrick R, Dharmadhikari S, et al. Synthetic Peptide CK2.3 Enhances Bone Mineral Density in Senile Mice. J Bone Res. 2018;6(2). doi: 10.4172/2572-4916.1000190.
6. Mubbunu L, Bowa K, Petrenko V, Silitongo M. Correlation of Internal Organ Weights with Body Weight and Body Height in Normal Adult Zambians: A Case Study of Ndola Teaching Hospital. Anat Res Int. 2018;2018:4687538. PubMed PMID: 29850249.
7. Moseychuk O, Akkiraju H, Dutta J, D'Angelo A, Bragdon B, Duncan RL, et al. Inhibition of CK2 binding to BMPRIa induces C2C12 differentiation into osteoblasts and adipocytes. J Cell Commun Signal. 2013;7(4):265-78. doi: 10.1007/s12079-013-0199-1. PubMed PMID: 23637019; PubMed Central PMCID: PMCPMC3889251.
8. Weidner H, Yuan Gao V, Dibert D, McTague S, Eskander M, Duncan R, et al. CK2.3, a Mimetic Peptide of the BMP Type I Receptor, Increases Activity in Osteoblasts over BMP2. Int J Mol Sci. 2019;20(23). PubMed PMID: 31771161.
9. Vrathasha V, Weidner H, Nohe A. Mechanism of CK2.3, a Novel Mimetic Peptide of Bone Morphogenetic Protein Receptor Type IA, Mediated Osteogenesis. Int J Mol Sci. 2019;20(10). Epub 2019/05/24. doi: 10.3390/ijms20102500. PubMed PMID: 31117181.
10. Jaroma AV, Soininvaara TA, Kröger H. Effect of one-year post-operative alendronate treatment on periprosthetic bone after total knee arthroplasty. A seven-year randomised controlled trial of 26 patients. Bone Joint J. 2015;97-b(3):337-45. Epub 2015/03/05. doi: 10.1302/0301-620x.97b3.33643. PubMed PMID: 25737517.
11. Turner AS, Mallinckrodt CH, Alvis MR, Bryant HU. Dose-response effects of estradiol implants on bone mineral density in ovariectomized ewes. Bone. 1995;17(4 Suppl):421s-7s. Epub 1995/10/01. doi: 10.1016/8756-3282(95)00321-4. PubMed PMID: 8579947.
12. Iwaniec UT, Samnegård E, Cullen DM, Kimmel DB. Maintenance of cancellous bone in ovariectomized, human parathyroid hormone [hPTH(1-84)]-treated rats by estrogen, risedronate, or reduced hPTH. Bone. 2001;29(4):352-60. Epub 2001/10/12. doi: 10.1016/s8756-3282(01)00582-8. PubMed PMID: 11595618.
13. Fuse H, Fukumoto S, Sone H, Miyata Y, Saito T, Nakayama K, et al. A new synthetic steroid, osaterone acetate (TZP-4238), increases cortical bone mass and strength by enhancing bone formation in ovariectomized rats. J Bone Miner Res. 1997;12(4):590-7. Epub 1997/04/01. doi: 10.1359/jbmr.1997.12.4.590. PubMed PMID: 9101370.

Reviewer 2 Report

Introduction

  1. The term "systematically injected" was used in several places in the manuscript.  Please confirm if you mean "systemically" or "systematically"?
  2. It was argued in the introduction that the SD rat osteoporosis model has the advantage of validated increase in osteoclast surface and bone turnover, however the manuscript did not report this part.
  3. Although a lot of previous work has been done on the novel peptide CK2.3 including some mechanistic pathways involved.  However, a lot of these relevant assessments have not been performed (or reported) on this animal model.  A clear hypothesis is lacking by the end of the introduction making this paper read like a simple observational study.

Results

  1. The presentation of results seems to not follow any logical order.  It was bone microarchitecture, BMD, mechanical property, BMD again, and then organ.
  2. The annotations of a, b, c above each bar is better explained in the legend of each figure.
  3. The presentation of microCT or BMD data across various time-points on separate graphs may not be the best to demonstrate the evolution of BMD or microarchitecture.
  4. One important aspect of this paper is the successful establishment of the osteoporotic animal model.  the baseline measurement of the microarchitecture should also be shown.  Since each animal received assessment repeatedly over 3 time-points.  You may also consider using the repeated measures ANOVA to investigate the differences over time in each of the parameters reported.
  5. To demonstrate the effect of OVX and the treatment effect of the novel compound, atrophy of the uterus should also be presented.
  6. There should also be 3 groups of animals for the trabecular BMD at week 0?  The data seems to shown no change of BMD over the 12 week period in the OVX group.

Methods

  1. At what age were OVX operation performed?  Were the baseline measurements by microCT or SPA taken before the OVX operation or after?
  2. Was baseline measurement also taken before or after the OVX in all animals?

Author Response

We appreciated the opportunity to revise the manuscript for reconsideration. We also want to thank the reviewer for the thoughtful and thoroughly comments on the manuscript. We took every comment seriously and revised the manuscript accordingly. The following content is a point-by-point revision based on reviewer’ comments and suggestions

Reviewer 2
Introduction
1. The term “systematically injected” was used in several places in the manuscript. Please confirm if you mean systemically” or “systematically”?
I appreciate that the reviewer pointed out the mistake in the manuscript. We meant to use the word “systemically”. We fixed the word throughout the manuscript.
2. It was argued in the introduction that the SD rat osteoporosis model has the advantage of validated increase in osteoclast surface and bone turnover, however, the manuscript did not report this part.
The mention of SD rat osteoporosis model has the advantage of validated increase in osteoclast surface and bone turnover was to explain the choice of OVX SD rats rather than OVX mice in this study. The effect of CK2.3 inhibited osteoclast surface was reported in our previous study in 6-month-old retired breeder mice(1). However, 6-month-old retired breeder mice didn’t mimic the post-menopausal osteoporosis in women. Thus, in this study we chose OVX rat model.
3. Although a lot of previous work has been done on the novel peptide CK2.3 including some mechanistic pathways involved. However, a lot of these assessment have not been performed (or reported) on this animal model. A clear hypothesis is lacking by the end of the end of the introduction making this paper read like a simple observational study.
We understood the concern of the review about the nature of this study. While it read like an observational study, the important of this study was to demonstrate the effect of CK2.3 in a standard osteoporosis model. Additionally, the response to bisphosphonates, parathyroid hormones, selective estrogen receptor modulator, and calcitonin in post-menopausal women was successfully predicted by OVX rat model (2). Thus, the results in this study could be used to predict the response of post-menopausal women to CK2.3. It would allow us to safely plan our future clinical studies.
We also included a hypothesis of the study in the Introduction.
On page 3 line 156-158 “A hypothesis of this study was that systemic injection of CK2.3 increased the BMD and bone architecture in OVX rats without influencing internal organs”

Results
1. The presentation of results seems to not follow any logical order. It was bone microarchitecture, BMD, mechanical property, BMD again, and then organ.
We acknowledged the concern of the reviewer about the logical order of the results. In fact, we ordered the results based on organ classification such as femur, spine, and organ. However, in agreement with the reviewer, we reordered them bone microarchitecture, BMD, mechanical property, and organ.
2. The annotations of a, b, c above each bar is better explained in the legend of each figure.
We agreed with the reviewer to clarify the annotations a, b, c in each graph. An explanation of the annotations was added in the legend of each figure. “(a) indicated a significant difference over SHAM, (b) indicated a significant difference over OVX or OVX+PBS, (c) indicated a significant difference over OVX+CK2.3.”
3. The presentation of microCT or BMD data across various time-points on separate graphs may not be the best to demonstrate the evolution of BMD or microarchitecture.
We understood the concern of the reviewer about this issue. The decision to separate the graphs was because we missed the opportunity to measure the baseline (week 0) of microCT. We first intended to only measure the femur BMD on live rats, and microCT would be performed after they had been sacrificed. However, we realized that it would put measurement of femur BMD and microarchitecture into two different methodologies. Thus, we decided to measure the microarchitecture by microCT on live rats as well. However, since we had already missed the opportunity to measure the baseline of microarchitecture, combining microCT and BMD graphs might result in a presentation that might cause confusion to readers.
4. One important aspect of this paper is the successful establishment of the osteoporotic animal model. The baseline measurement of the microarchitecture should also be shown. Since each animal received assessment repeatedly over 3 time-points. You may also consider using the repeated measures ANOVA to investigate the differences over time in each of the parameters reported.
We agreed with the reviewer that including baseline measurement of the microCT would make the paper stronger. However, we missed the opportunity to measure the baseline of microCT. Additionally, with only two time points (week 4 and week 12), the requirement for repeated measure ANOVA wasn’t met.
5. To demonstrate the effect of OVX and the treatment effect of the novel compound, atrophy of the uterus should also be presented.
We were aware of this issue. We decided not to perform atrophy of the uterus to demonstrate the effect of OVX on rats because the effect of OVX on rats had already been published elsewhere. As indicated in the Introduction, “OVX rats are an excellent model that emulates the estrogen depletion that leads to OP in post-menopausal women [30-32]… ovariectomy-induced bone loss in outbred Sprague Dawley rats was validated by the increase of osteoclast surface and bone turnover [36,37]”. Furthermore, a personal communication with the vendor, Charles River, confirmed the operation was a success “Charles River performs the surgery (ovx) and verifies complete removal by a second impartial person who examines the removed ovary to make sure that it is a total and complete ovary.” Thus, we opted out of demonstrating the effect of OVX on the rats in this study.
Atrophy of the uterus in osteoporosis is in part caused by post-menopausal estrogen deficiency. Our novel compound, CK2.3, was an activator of BMP2 signaling transduction (3-5). Thus, we didn’t expect CK2.3 to alleviate the estrogen deficiency. We extrapolated that CK2.3 wouldn’t lessen the atrophy of the uterus.
6. There should also be 3 groups of animals for the trabecular BMD at week 0? The data seems to show no change of BMD over the 12 week period in the OVX group.
We understood why the reviewer was confuse about the missing animal group in week 0. However, there wasn’t a missing animal group in week 0. We regretted to cause confused with the labeling. Week 0 was the baseline measurement before treatment (PBS or CK2.3) was administrated. Thus, it had only SHAM operated and OVX operated. To fix this confusion, we clarified it in the figure legend.
On page 4 line 203-204 “Figure 2. CK2.3 increased trabecular BMD femoral head of OVX rats. (A) OVX rats had lower baseline (week 0) BMD than SHAM rats…”
After the OVX surgery, the rats were rested for 4 weeks before treatment. The measurements at week 4 and week 12 post-treatment were week 8 and week 16 post-OVX surgery. Studies have indicated there isn’t significant difference in femur BMD between week 8 and week 16 post-OVX surgery (6, 7).

Methods
1. At what age were OVX operation performed? Were the baseline measurements by microCT or SPA taken before the OVX operation or after?
The rats were 15-week-old when OVX operation were performed. The age could be found in the Methods “Fourteen female OVX and seven placebo (SHAM) operated Sprague Dawley rats, 15 weeks of age (105 days) were purchased from Charles River (QTE# 20150209).”
As mentioned above, we missed the opportunity to perform the baseline measurements by microCT. We, however, were only able to measure baseline measurement for femur BMD, and this was done 4 weeks after OVX operation. We purchased the rats from Charles River and asked them to perform OVX operation before shipping. This was also indicated in the Methods “Four weeks after arrival, baseline bone mineral densities (BMD) of the trabecular left femur were measured using the Bruker SkyScan 1276 X-ray micro-computed tomography (micro-CT).”
2. Was baseline measurement also taken before or after the OVX in all animals?
As the rats were OVX operated before they were shipped to us, the baseline measurement of femur BMD was taken 4 weeks after the OVX in all animals. It was indicated in the Methods “Four weeks after arrival, baseline bone mineral densities (BMD) of the trabecular left femur were measured using the Bruker SkyScan 1276 X-ray micro-computed tomography (micro-CT).”

References
1. Nguyen J, Weidner H, Schell LM, Sequeira L, Kabrick R, Dharmadhikari S, et al. Synthetic Peptide CK2.3 Enhances Bone Mineral Density in Senile Mice. J Bone Res. 2018;6(2). doi: 10.4172/2572-4916.1000190.
2. Iwaniec UT, Yuan D, Power RA, Wronski TJ. Strain-dependent variations in the response of cancellous bone to ovariectomy in mice. J Bone Miner Res. 2006;21(7):1068-74. Epub 2006/07/04. doi: 10.1359/jbmr.060402. PubMed PMID: 16813527.
3. Bragdon B, Thinakaran S, Moseychuk O, King D, Young K, Litchfield DW, et al. Casein kinase 2 beta-subunit is a regulator of bone morphogenetic protein 2 signaling. Biophys J. 2010;99(3):897-904. Epub 2010/08/05. doi: 10.1016/j.bpj.2010.04.070. PubMed PMID: 20682268; PubMed Central PMCID: PMCPMC2913199.
4. Bragdon B, Thinakaran S, Moseychuk O, Gurski L, Bonor J, Price C, et al. Casein kinase 2 regulates in vivo bone formation through its interaction with bone morphogenetic protein receptor type Ia. Bone. 2011;49(5):944-54. Epub 2011/07/19. doi: 10.1016/j.bone.2011.06.037. PubMed PMID: 21763800.
5. Bragdon B, Bonor J, Shultz KL, Beamer WG, Rosen CJ, Nohe A. Bone morphogenetic protein receptor type Ia localization causes increased BMP2 signaling in mice exhibiting increased peak bone mass phenotype. J Cell Physiol. 2012;227(7):2870-9. Epub 2011/12/16. doi: 10.1002/jcp.23028. PubMed PMID: 22170575; PubMed Central PMCID: PMCPMC3309108.
6. Lee C, Lee JH, Han SS, Kim YH, Choi YJ, Jeon KJ, et al. Site-specific and time-course changes of postmenopausal osteoporosis in rat mandible: comparative study with femur. Sci Rep. 2019;9(1):14155. Epub 2019/10/04. doi: 10.1038/s41598-019-50554-w. PubMed PMID: 31578360; PubMed Central PMCID: PMCPMC6775083.
7. Lei Z, Xiaoying Z, Xingguo L. Ovariectomy-associated changes in bone mineral density and bone marrow haematopoiesis in rats. International Journal of Experimental Pathology. 2009;90(5):512-9.

Round 2

Reviewer 1 Report

The manuscript has been revised well. I think this manuscript will be acceptable.

Reviewer 2 Report

Limited to the design of the experiment, most of the previously raise concerns cannot be addressed at this time, which I think would affect the strength of the evidence.

Round 3

Reviewer 2 Report

I think the limitations have been clearly supplemented to the paper.  I have no further suggestions and would like to congratulate the authors for their work.